# Profitability and Cost Analysis for Contract Broiler Production in Turkey

**DOI:** 10.3390/ani13132072

**Published:** 2023-06-22

**Authors:** Suleyman Karaman, Yavuz Taşcıoğlu, Osman Doğan Bulut

**Affiliations:** 1Faculty of Agriculture, Department of Agricultural Economics, Akdeniz University, Antalya 07058, Turkey; ytascioglu@akdeniz.edu.tr; 2Faculty of Agriculture, Department of Agricultural Economics, Iğdır University, Iğdır 76000, Turkey; odogan.bulut@igdir.edu.tr

**Keywords:** costs, contract production, net enterprise income

## Abstract

**Simple Summary:**

This study aims to determine the income level of contracted broiler breeders in terms of earnings and to calculate the profitability, cost, and return of contracted production. In broiler production, the contracted production model between the breeder and the integrated broiler business that supplies the product to the market is applied in Turkey as well as in the world. Broiler production is completed in 5–7 weeks in the normal process. This process may vary depending on slaughter weight, market conditions, carcass broiler weight demands of consumers, feed prices, broiler sales price, and operating conditions. However, in some periods, integrated broiler enterprises make the broiler slaughter earlier in order not to experience disruptions in product supply in the supply chain. In such cases, to determine whether broiler breeders suffer from income loss, breeding parameters were calculated as cost and net income gain in the study. The results show that the entrepreneurs generated a net income of $9.197 per m^2^ and $0.541 per broiler during the rearing period from broiler breeding. Despite the low earnings in contracted broiler breeding, it is continued because it allows the on-site evaluation of the workforce of family businesses engaged in breeding. Integrated broiler enterprises can procure products regularly throughout the year through contracted production.

**Abstract:**

This study uses the data obtained from 63 broiler farms engaged in contract farming in Akhisar, Turkey. The average feed conversion ratio in the broiler farms is 1.75, the average live weight 2.25 kg, and the mean market age 38.9 days. The feed conversion rate and the income generated are highly correlated (r = −0.76). The production index is 313.4. According to this production index value, 47.6% of the enterprises are below the average production index. It is highest when the marketing age is greater than 38 days, less than and equal to 40. In other words, it is the optimum market age range where carcass yield is at maximum. The average mortality rate is 4.68%. In 52.4% of the enterprises, the mortality rate is above 5%. There is a low level of correlation between the mortality rate and income (r = −0.26). In broiler farming, mortality rate, and feed conversion ratio are factors that directly affect the success and income of the breeder. In broiler farming, the heating cost has the largest share of the total cost, followed by the litter cost. They are followed by labor, electricity, and pesticide costs. Variable costs account for three-quarters of the total operating costs. The farms have a mean gross value of production of $23.797 per m^2^ and $1.400 per broiler in a breeding period. The profit margin is 0.572 $/kg per broiler. The mean enterprise net income in the breeding period is $9.197 per m^2^ and $0.541 per broiler. These findings suggest that broiler farming is a profitable venture.

## 1. Introduction

The broiler industry in Turkey is fully vertically integrated with contract production. It is structured to include all processes, from rearing to the marketing of meat. Broiler production contracts are legal agreements between integrators and producers that are binding on producers. Contracts are concluded to organize broiler production. They set out the terms of the production relationship between the breeders and the integrators [1]. These conditions may differ between integrated broiler companies [2]. There are methods to specify the responsibilities of integrators for the provision of inputs and the payments to breeders when drawing up broiler production contracts. Integrators provide one-day-old chicks, feed, medicine, healthcare services, and extension services free of charge. Costs such as poultry house, litter (sawdust/paddy), heating (coal/gas), electricity, labor (care), water, disinfection, and operation, maintenance, modification, and renovation expenses of the poultry house are borne by the breeder [3].

Integrators benefit breeders by providing them with the opportunity to earn income with relatively low capital. In addition, they offer alternative opportunities for breeders to overcome their capital constraints. For example, fuel and litter that cannot be supplied by the breeder are supplied by the integrator if the breeder requests the integrator at the beginning of the production period to do so, and the request is approved by the integrator, and the expenses incurred are deducted from the breeder’s contract production income at the end of the period [4].

Contracting in broiler production allows the risk to be passed on from breeders to integrators. This risk shift is due to the transfer of some of the production risks, most notably the price risk [5]. Contracting has been shown to be an effective tool for breeders to avoid risk. A broiler production contract can effectively protect the interests of breeders in the event of a market or production shock [6]. It was found that risk-averse breeders are more likely to enter into production contracts and less likely to adopt new technologies [7]. Integrators resolve any problems faced by breeders during the contract period and can handle the alterations in technology. During each breeding period, authorized personnel of the integrator visit the poultry houses at least five times to observe the conditions of the breeder and the conditions of the broilers, including their health, nutrition, heating, ventilation, and live weight gain, as well as catering for their needs such as a vaccine, medicine, etc. During the visits, the current state of production is assessed, providing recommendations to increase productivity and measuring the performance of the breeder. On the other hand, the breeder agrees to indemnify the company’s losses arising out of damages incurred by the contract production inputs and/or broilers belonging to the integrator in his poultry house due to fire, disease, environmental conditions, poor maintenance, and similar reasons [8]. 

Broiler production contracts are often criticized by breeders. They complain about the earnings resulting from the contract terms prepared by integrators. They claim that they actually suffer a small number of negative income effects due to their broiler production contract. Production contracts are drawn up with appropriate incentives for breeders to manage their broiler farming business in a way that maximizes the profits of integrators. At the same time, significant rewards are granted to attract new breeders and to ensure the continuity of business with the existing breeders. This pushes breeders to attempt to maximize their net operating income within contractual constraints. Assuming that an excellent incentive mechanism or technical support mechanism exists, both breeders and integrators will maximize their profits. In this context, the production performance and profitability that contract breeders achieve based on their knowledge and experience, as well as the technical support they receive, needs to be investigated, along with the poor level of income generated, which both breeders and integrators claim to suffer. 

The present study is intended to identify the descriptive values of contract production measurements and to calculate the profitability, costs, and proceeds of contract production in order to determine the income level of breeders. It also aims to evaluate the effects of different market ages, which are decided by integrators, on performance parameters. In addition, it discusses what needs to be done to improve the income of contract breeders. 

## 2. Materials and Methods

### 2.1. Data Collection 

This study was carried out on 63 farms engaged in contract broiler production in Akhisar, Turkey. The farms were selected purposefully. It is assumed that enterprises with different poultry house capacities can represent the population. Every enterprise has its own poultry house. The costs and labor data for the breeding period were obtained from their records through direct interviews. Other data were obtained from the flock productivity database managed by the integrator. This database provides information on management factors and records for each flock. The flock data are for the breeding period from April to May. Production performance parameters were calculated using the flock productivity database. A profitability analysis was conducted using the data from the flock productivity database and the records of inputs maintained by enterprise managers. 

### 2.2. Data Analysis 

In the study, the contract broiler production performance of the enterprises was measured in flocks. Broiler production performance can be measured using various parameters, including livability, mean daily live weight gain, and feed conversion ratio (FCR). The production index and target FCR systems based on the live weight determined by the integrator are used to evaluate the broiler production performance. The production index (European Production Efficiency Factor, EPEF) is estimated using livability, average daily weight gain, and feed conversion ratio. The production index = [(livability × average daily weight gain)/feed conversion ratio] × 100. The production index gives information about the carcass yield performance of the enterprises. Livability = (number of live broilers sent to slaughter/number of broilers in the poultry house) × 100; Average daily weight gain = (average carcass weight/slaughter age); Feed conversion ratio = (total feed given/total live weight) [9].

Market age differs among enterprises. The flocks were grouped for analysis by market age. The market age groups are MA1 (≤38), MA2 (38 < MA2 ≤ 40), and MA3 (MA3 > 40 days). The parameters used to evaluate life performance and economy include flock average market weight, livability, average daily weight gain, feed conversion ratio, production index, feed consumed (kg/bird), production cost ($/kg), and feed conservation ratio. In addition, a one-way analysis of variance was used to test whether broiler production parameters differed according to marketing age groups. Linear regression was used to evaluate the effect of market age on FCR and average flock market weight.

Variable costs of broiler production are the costs that increase or decrease depending on poultry house capacity. Among the variable cost items, the costs of chicks, feed, some vaccination, veterinarian, transportation, etc., are borne by the integrator. Other variable costs such as disinfection and cleaning, labor, heating, water and water analysis, electricity and lighting, equipment repair and maintenance, sawdust and litter, medicine, and insurance are fully borne by breeders. In addition, the revolving fund interest calculated for the inputs supplied by the breeders themselves during the breeding period is also included in variable costs.

To calculate the depreciation for the building and equipment, the straight-line method was used, taking into account the economic life of the fixed assets. The depreciation of equipment was calculated after asking the value of the new equipment to the manufacturers and determining its potential economic life [10]. The calculated annual depreciation values were divided into 6.5 breeding periods.

Profitability is measured using net operating income. The best indicator for determining the economic performance of a business is net operating income [11]. It is the return on family labor, capital owned and managed. It is defined as the difference between revenues and costs, excluding family labor costs [12].

## 3. Results

### 3.1. Descriptive Statistics 

The distribution of broiler enterprises by their poultry house chick capacity is given in Table 1. The poultry house chick capacities and floor areas (m^2^) of the enterprises were obtained from the integrator database records. 47.6% of the enterprises have a poultry house capacity in the range of 10,000–20,000 birds. Only 7 enterprises have a poultry house capacity of over 40,000 birds. The poultry house capacity of the enterprises is directly related to their floor area. The integrator supplies 17 chicks per square meter. Premiums are paid by the integrator according to size groups for the poultry house areas. As the poultry house area increases, the premium per square meter also increases. 

The distribution of breeders by years of experience is given in Table 2. The breeders’ years of experience were retrieved from the integrator’s database. The experience of the breeders significantly affects the production performance and profitability of the enterprises. One-fourth of the enterprises do not have experience. The number of enterprises with 10 or more years of experience is 11. The integrator pays the breeders a premium based on their length of experience in years. As the length of experience increases, the premium value also increases.

Performance parameters of contract broiler production are presented in Table 3. The mean market age of broilers is 38.9 days. The lowest market age among the enterprises during the breeding period is 33 days. The mean live weight is 2.25 kg. The highest mean live weight in individual farms is 2.61 kg. The livability is 94.3% on average. The feed conversion ratio is 1.75 on average. The lowest feed conversion ratio among the enterprises is 1.61, while the highest is 1.97. The production index is 313.4. 

The production index is below the average in 47.6% of the enterprises. An average of 3.94 kg of feed is consumed per chicken. The lowest feed consumption per chicken among the enterprises is 2.74 kg. The average mortality rate is 4.68%. The highest mortality rate among enterprises is 7.91. The mortality premium varies depending on whether it is below or above 5%. In 52.4% of the enterprises, the mortality rate is above 5%. In broiler production, mortality is a factor that directly affects the success and income of the breeder

### 3.2. The Effect of Broiler Market Age on Performance Parameters

Since broilers are the property of the integrator in contract production, they can be sent to slaughter at any time by the management, depending on the market demand, regardless of their age and weight. Breeders do not have the right to challenge any decision to do so or to refrain from delivering the broilers pursuant to the contract. Hence, broiler market ages differ among enterprises. According to the One-way Analysis of Variance (ANOVA), there is a significant difference between marketing age groups at a 1% level of significance. In this respect, the ANOVA test shows that average marketing weight, feed conversion ratio, and feed consumed (kg/bird) parameters show a significant difference at a 1% significance level according to marketing age groups. On the other hand, the ANOVA test, livability (%), production index, and feed conservation ratio bonus parameters do not show a significant difference according to marketing age groups (Table 4).

Wang et al. (2014) note that the highest EPEF value gives the optimum return, and the best slaughter age is the day on which the highest EPEF and the lowest FCR are recorded [6]. The production index, which is an indicator of the general production profile, indicates the broiler carcass yield performance. The production index is the highest in MA2. In other words, the optimum market age in terms of carcass yield is MA2 (38 < MA2 ≤ 40). The market age has an impact on the producer’s income, as the integrator pays premiums according to the target FCR system determined by live weight. A comparison of the market age (MA) ranges indicates that the highest premium per kg is paid for broilers in MA2. Accordingly, the highest income per unit area was generated from broilers in MA2. The livability was highest in MA2. Significant differences between MA groups affect the economic performance of the producer (Table 4).

Linear regression analysis was performed to determine the effect of broiler marketing age on average live weight, feed intake per broiler, and feed conversion ratio (Table 5). These three parameters show significant differences according to marketing age groups (Table 4). According to the linear regression result, it shows that when marketing age increases by one day, weight can increase by 0.086 kg. Depending on the FCR, the average weight increases gradually with age. In broiler breeding, a high FCR level increases the production cost. In this respect, a low feed conversion ratio is desirable. The linear regression result shows that when the marketing age increases by one day, the FCR may increase by 0.016% on average. In broiler breeding, the FCR level increases as the marketing age increases (Table 5).

High marketing age is not desirable in terms of feed consumption per broiler. According to the linear regression result, as the marketing age increases by one day, the amount of feed consumed per broiler increases by 0.183 kg (Table 5). It is also observed that feed consumption per broiler increases in marketing age groups. In this respect, the economic loss threshold should be taken into consideration for broiler breeding to be sustainable.

### 3.3. Correlation Analysis between the Factors of Total Return and Bonus Paid

At the end of the production period, the breeder’s performance results are communicated by the integrator. These results are compared with the performance indicators (feed conversion ratio, loss rate, live weight, slaughter age, etc.) determined by the integrator. The broilers that died en route to the slaughterhouse are deducted from the delivery amount and live weight. Performance payment is based on a fixed payment per kg of live weight delivered. In addition, a fixed amount of fuel per kg of live weight is supplied by the integrator. Performance-based premium payment consists of FCR premium (±), mortality premium (±), km premium (±), m^2^ premium, and seniority premium. The live weight kg fee of the breeder is determined according to the fixed base price, fixed fuel subsidy price, FCR premium, km premium (±), m^2^ premium, and seniority premium.

The lowest and highest live weight payments to the breeders were $0.15 and $0.24, respectively. Live weight payment to the breeders was $0.2 on average. 38.9% of the enterprises earned a live weight fee below $0.2, and 61.9% earned above $0.2.

Pearson correlation analysis was conducted to determine the direction and extent of the relationship between the live weight fees earned by the breeders and the factors for which a premium is paid. There is a low positive correlation between the breeder’s length of experience and the fees earned at a 5% significance level. It is expected that the effect of the experience of the breeder on the live weight obtained is always positive. There is a low negative correlation between the distance of the poultry house to the integrator’s slaughterhouse and the fees earned at a 1% level of significance. The greater the distance, the lower the premium received. There is a low negative correlation between mortality rate and fees earned at a 5% level of significance. The lower the mortality rate, the higher the fees earned. There is a very high negative correlation between the feed conversion ratio and fees earned at a 1% level of significance (Table 6). It is desirable to have a low feed conversion ratio value in broiler breeding. In other words, it is ensured that the cost of feed, which is the most important input in broiler breeding, decreases and the income obtained increases.

### 3.4. Broiler Production Costs and Net Enterprise Income 

Disinfection and cleaning costs include the expenses for disinfectants used in house washing, spraying, and rodent control expenses. Disinfection and cleaning costs are $0.366 per m^2^ and $0.022 per broiler. The development and health of broilers are regularly checked by the company’s veterinarians. The drugs used by the breeders during the contract production are supplied by the integrator at the cost of the breeders. The prices paid for the drugs used by the breeders are included in the breeder database records of the integrator. The integrator deducts a medicine cost of $1.457 per m^2^ and $0.086 per broiler from the account of the breeders. Medicine costs rank fifth among cost items. Temporary labor is used for disinfection, spraying, and other work. Temporary worker wage paid by the enterprises is $0.555 per m^2^ and $0.033 per broiler (Table 7). 

Water costs are fully borne by the breeders. For water analysis, the breeders pay $0.246 per m^2^ and $0.014 per broiler. For the volume of water consumed, the breeders incur an expense of $0.077 per m^2^ and $0.005 per broiler. Electricity expenses incurred by the breeders before, during, and/or after contract production are borne by themselves. The breeders incur an electricity cost of $1.857 per m^2^ and $0.109 per broiler (Table 7). In broiler production, electricity cost ranks fourth among all cost items. It was found that $0.941 per m^2^ and $0.055 per broiler were incurred for the generator, feeder, drinker, ventilation systems, and other repair and maintenance operations during the production period. The breeders prepare their poultry house according to the integrator’s production program and as requested by the technical staff by supplying the necessary and sufficient amount of litter material (sawdust, paddy husk, straw, etc.) for the coops at their own cost. The breeders spend $2.026 per m^2^ and $0.119 per broiler on sawdust and litter. Sawdust and litter costs in broiler production rank second place among cost items (Table 7). 

The breeders heat their poultry houses with coal stoves (wood and coal) or with liquid petroleum gas. Heating costs are fully borne by the breeders. The breeders spend $3.889 per m^2^ and $0.229 per broiler on heating. In contract broiler production, the heating cost has the highest value among cost items. 

Before each production period, the breeders have to ensure that their poultry house is insured against fire and other natural disasters and that the integrator appears as the payee in the insurance policy in case of any event, as a result of which the breeder will be entitled to compensation. The insurance premium is deducted from the income of the breeder at the end of the breeding period as an insurance cost of $0.474 per m^2^ and $0.028 per broiler. 

Fixed costs consist of administrative expenses, labor (family labor) costs, poultry house capital interest, building and equipment depreciation, and withholding tax. During the breeding period, the majority of the required labor is provided by employing the family members in the poultry house. The family labor costs during the breeding period are $1.972 per m^2^ and $0.116 per broiler. Family labor cost ranks third among all cost items. Building and equipment depreciation is $0.568 per m^2^ and $0.033 per broiler for the breeding period. Allocation of depreciation is obligatory for the renewal of the machinery and buildings whose economic life has expired or which need to be replaced due to new technological developments. 

The gross value of production is obtained by subtracting the expenses (insurance premium, water analysis cost, disinfection and cleaning cost, and mortality premium) from the production fee earned by the breeders. The income that the breeders generate is determined by deducting the withholding tax from the total income. For the breeding period, a gross value of production of $23.797 per m^2^ and $1.400 per broiler is obtained. In this breeding period, a gross income of $23.817 per m^2^ and $1.401 per broiler was generated. The profit margin is 0.572 $/kg per broiler. The mean enterprise net income in the breeding period is $9.197 per m^2^ and $0.541 per broiler (Table 7). These findings suggest that broiler production is a profitable venture.

## 4. Discussion

The study should also consider factors that will affect the sustainability of business production, such as feed conversion rate, production index, average mortality, and heating costs. In our study, the feed conversion ratio was found to be between 1.61 and 1.97. According to Patil et al., this value was calculated as 1.8 [13]. In the study conducted by Gholami et al., the FCR was found to be 1.84 in mild, humid, semi-arid, and alpine areas, and 1.86 in hot and dry areas [14]. These values show that both our study and the other enterprises are close, the feed conversion ratio of the enterprises is below the targeted feed conversion ratio, and they can receive FCR premium due to the low feed conversion ratio. However, in a study of broiler breeding in which different diet programs were applied, Liu et al. found that the feed conversion ratio in chickens was 1.445–1.511, and Melo et al. found it to be 2.00–2.45 [15,16]. This value may differ according to regions, production systems, and feed ingredients.

The broiler production index was found to be 313.4 in the study, and Sasaki et al. calculated it as 283.9 in the study conducted by [17]. On the other hand, Gholami et al. In the study conducted by the production index, it was calculated as 321.2 in mild and humid areas, 320.0 in semi-arid areas, 323.9 in alpine areas, and 314.0 in hot and dry areas [14]. With our study in terms of the value of the production index, a value close to the data showing the hot and dry climate characteristics of Gholami’s study was found [14]. This value reflects the climatic characteristics of the study area. It is seen that climatic characteristics are important in terms of production index in broiler production.

In the study, the average death rate of broiler farms was found to be 4.68%. This ratio is desired to be below 5% in enterprises. In the study conducted by Delabouglise et al. (2019), it was found that the average mortality rate per flock in broiler breeding on the basis of diseases was 19.9%, and 60% was due to the disease [18]. This value shows that the spread of diseases due to the characteristics of the breeding environment increases mortality rates. For this reason, it shows the necessity of taking into account the hygiene rules in broiler breeding. In a study conducted by Vieira et al. (2011) in Brazil, a mortality rate of 0.42% in summer and 0.28% in winter was reported [19]. It is seen that Brazil benefits from climatic advantages and breeding experiences in broiler production. Knezacek et al. (2010) reported a mortality rate of up to 1.4% in a study conducted in Canada [20]. In the study conducted by Grilli et al. (2018) in Central Italy and dividing the enterprises as large, medium, and small, the mortality rate was found to be 0.52%, 0.47%, and 0.31%, respectively [21]. In the study conducted by İkikat Tümer in Turkey, this rate was found to be as high as 9.68% [22]. The fact that this rate is low in the enterprises in the research region. Although it showed that the relevant legislations of the enterprises are applied for production, the heating is at the desired level, and the coop rest periods are followed, it showed that this value is higher in Turkey compared to the countries that have experience in broiler breeding.

In the study, heating costs per m^2^ of broiler breeding were calculated as $3.889 and $0.229 per broiler. Heidari et al. and Ertürk and Tatlıdil found heating costs to be among the most important inputs in studies [23,24]. In the study by Yeni and Dağdemir, it was explained that heating costs vary depending on regional climatic conditions and, therefore, high heating costs are the biggest factor contributing to negative net income [25]. The enterprises in the research area are in an advantageous position compared to the other researched regions due to climatic conditions. In this context, climate conditions, heat insulation [26], air conditioning, ventilation, etc., are in the investment processes of broiler production enterprises. Considering these factors, it can be seen as important for businesses both in terms of cost-reducing factors and business sustainability.

Contract production in broiler farming has many advantages and disadvantages. The most important advantages of contracted broiler breeding are the short production cycle, the purchase guarantee by the company, and the ease of credit and input supply. The broiler production capital cycle can be repeated 6–7 times a year [27]. Considering that the broiler production cycle is 6–7 weeks, the capital cycle in broiler production is very fast compared to the capital cycle in other animal production types due to its short production cycle and high economic return [28]. On the other hand, it is an important advantage to provide a purchase guarantee for the breeders by the integrated company in contracted broiler production. According to the contract, while the breeders are held responsible for broiler production, the companies are responsible for the marketing of the broilers. Thus, the risk of broiler marketing for breeders is eliminated. Therefore, a stable income is provided for broiler breeders.

Inspecting the production through technical field personnel at almost every stage of production [29], from when the birds are harvested to the placement of new flocks in the poultry houses, in the supply and distribution of inputs, especially chick and feed, to the breeder of the integrated broiler company, and fuel allowance in the winter months. Provides support in many ways, from modernizing production facilities to building and equipping the grower’s accommodation facilities within the framework of the provisions of the contract. On the other hand, the breeder produces by covering the land, broiler house, equipment, labor, and normal operating expenses [30]. This production, which is done in accordance with the mutual trust method, positively affects the performance of the grower [30].

However, broiler breeding has some disadvantages despite these advantages. Many studies in the literature have stated that contract broiler farming causes environmental degradation, longer hours of work by breeders, the use of breeders’ children as cheap labor, and income inequality [31,32,33,34] (Singh, 2002a, Singh, 2002b, Porter and Phillips-Howard, 1997, Little and Watts 1994).

Huang et al. (2018), in their study on Chinese broiler breeders, determined that small-scale breeders did not reach their goals of limited profitability and high welfare [35]. In this study, they claimed that broiler breeders earn limited income in some production periods due to the production contract, but in fact, they are exposed to a small amount of negative income in terms of marketing age. Yeni and Dağdemir (2011) found in their study that broiler breeders earned negative income, while in the study conducted by Memken and Bellemere (2019), contracted breeders earned 10% more income on average than non-contracted breeders [25,36]. Contract production, which is considered profitable for breeders in this study and previous studies in this field, is generally directly related to the content of the contract. In contracts, there are problems such as prices being determined by the buyer company, late payments to the manufacturer, the contracts being prepared only by the buyer companies, or the responsibility being placed on the manufacturer. For this reason, the contracts should be of a quality that will ensure the sustainability of production and should be arranged in a way that does not cause a loss of income for both parties.

## 5. Conclusions

The fact that the inputs for contracted broiler production are supplied by the integrator, that the breeders do not have solvency problems to procure such inputs, that they do not face any marketing problems, and that they can operate this business together with their family members, especially in their own villages, has made this industry attractive. Since broiler production is a risky venture with a very low-profit margin, enterprises can survive in the market if they carry out their activities in a contract production system. In this respect, both the breeders and the integrator tend to continue with the contract production model. 

In broiler production, the heating cost has the largest share in the total cost, while litter cost ranks second among cost items. They are followed by labor, electricity, and pesticide costs. Variable costs account for three-quarters of the total operating costs. Entrepreneur net income, which is breeder income, shows that broiler production is a profitable venture. Since the contract drawn up by the integrator does not provide the breeders with the right to challenge the production processes, they can suffer a loss of income. In this context, the contracts should be prepared or prepared with the support of producer organizations, non-governmental organizations, and relevant ministries that have a say in the rural structure, and they should not only provide sectoral development but also avoid negativities in the consumption dimension and the functioning of the supply chain. Depending on the poultry house capacity, broiler production can be the main source of income for the family or provide the breeders with a side income and lucrative employment throughout the year.

## Figures and Tables

**Table 1 animals-13-02072-t001:** Number of broiler chickens per production cycle in the farms.

Number of Birds	Number of Farms	Percentage
<10,000	6	9.52
10,000–20,000	30	47.62
20,000–40,000	20	31.75
40,000 and above	7	11.20
Total	63	100.00

**Table 2 animals-13-02072-t002:** Broiler producers’ length of experience in years.

Years of Experience	Number of Farms	Percentage
0	16	25.40
1–2	9	14.29
2–3	4	6.35
4–5	9	14.29
6–10	14	22.22
10 and above	11	17.46
Total	63	100.00

**Table 3 animals-13-02072-t003:** Performance parameters of broiler chickens.

Parameters	Mean ± SD	Min	Max	CV (%)
Average market age (days)	38.9 ± 1.58	33	42	4.07
Average weight (kg)	2.25 ± 0.20	1.67	2.61	8.92
Livability (%)	94.27 ± 2.63	83.90	98.90	2.79
Feed conversion ratio	1.75 ± 0.07	1.61	1.97	4.15
Production index	313.37 ± 28.77	261	376	9.18
Feed consumed (kg/bird)	3.94 ± 0.38	2.74	4.53	9.62
Mortality (%)	4.68 ± 1.58	0.05	7.91	33.83

**Table 4 animals-13-02072-t004:** Evaluated parameters as a function of market age.

Parameter	MA1	MA2	MA3	SEM ^1^	*p*-Value
Average market age (days)	36.0 ^B^	39.1 ^AB^	40.7 ^A^	0.125	0.000 *
Average weight (kg)	1.976 ^B^	2.274 ^A^	2.346 ^A^	0.026	0.000 *
Livability (%)	94.40	94.62	93.06	0.404	0.160
Feed conservation ratio (FCR)	1.696 ^B^	1.753 ^AB^	1.801 ^A^	0.010	0.004 *
Production index (EPEF)	309	318	302	4.447	0.240
Feed consumed (kg/bird)	3.35 ^B^	3.98 ^AB^	4.22 ^A^	0.044	0.000 *
Feed conservation ratio bonus ($)	0.079	0.088	0.075	0.005	0.281

Note: MA1 (≤38), MA2 (38 < MA2 ≤40), MA3 (MA3 > 40 days); ^1^ Standard error of the mean; Different superscript letters in the same row indicate significant differences (B < A); * indicates 1% significance level.

**Table 5 animals-13-02072-t005:** Regression for selected parameters as a function of market age.

Dependent Variable	Feed Consumed (kg/bird)	Average Live Weight (kg)	FCR
Exploratory Variable	Coefficient	Coefficient	Coefficient
MA	0.183	0.086	0.016
	(0.020) *	(0.012) *	(0.006) *
C	−3.172	−1.103	1.136
	(0.772) *	(0.463) **	(0.215) *
R-squared	0.582	0.462	0.120
F-statistic	84.933	52.292	8.277
Prob (F-statistic)	0.000	0.000	0.006

Note: * and ** indicate 1% and 5% significance levels. The values in parentheses are the *p*-values.

**Table 6 animals-13-02072-t006:** Correlation analysis.

	Experience	Fees	Distance	Size	Mortality Rate	FCR
Experience	1.000					

Fees	0.265	1.000				
	(0.036)					
Distance	0.170	−0.386	1.000			
	(0.182)	(0.002)				
Size	−0.022	−0.072	−0.186	1.000		
	(0.866)	(0.575)	(0.144)			
Mortality rate	−0.115	−0.251	−0.043	0.007	1.000	
	(0.368)	(0.047)	(0.738)	(0.957)		
FCR	−0.058	−0.765	0.047	0.219	0.283	1.000
	(0.653)	(0.000)	(0.711)	(0.851)	(0.240)	

Note: The values in parentheses are the *p* values.

**Table 7 animals-13-02072-t007:** Broiler production costs and net enterprise income in a period.

Parameters	17 Birds/m^2^	Per Bird	Rank	%
Disinfection and cleaning	$0.366	0.022	11	2.39
Water charges	$0.077	0.005	16	0.50
Electricity and lighting	$1.857	0.109	4	12.12
Heating	$3.889	0.229	1	25.39
Litter	$2.026	0.119	2	13.23
Hired labor	$0.555	0.033	8	3.62
Water analysis fee	$0.246	0.014	12	1.61
Medication	$1.457	0.086	5	9.51
Insurance	$0.474	0.028	9	3.09
Equipment repair and maintenance	$0.941	0.055	6	6.14
Interest on operating expenses	$0.201	0.012	13	1.31
Total variable cost (1)	$12.089	0.712		
Housing capital interest	$0.174	0.010	14	1.14
Labor (Family) charges	$1.972	0.116	3	12.88
Withholding tax	$0.105	0.006	15	0.69
Depreciation of building and equipment	$0.568	0.033	7	3.71
General Administration Expenses	$0.408	0.024	10	2.66
Total fixed cost (2)	$3.227	0.189		
Total cost (1 + 2)	$15.316	0.901		100.00
Gross value of production (3)	$23.797	1.400		
Gross margin (3 − 1)	$9.736	0.572		
Gross return	$23.817	1.401		
Enterprise income	$11.730	0.690		
Net enterprise income	$9.197	0.541		

## Data Availability

The data may be obtained upon request from the authors.

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
