# Peer review of "Profitability and Cost Analysis for Contract Broiler Production in Turkey"

_animals, 2023, doi:10.3390/ani13132072_

Round 1

Reviewer 1 Report

The sections Material and Methods, Results and Discussion in the document are confused, authors should correct the document. See file attached

There are minor errors in english redaction. See file attached

Author Response

Dear reviewer
Our article has been revised in line with your suggestions.

Best Regards.

Reviewer 2 Report

The study entitled “Profitability and cost analysis for contract broiler production in Turkey” presents a production and financial analysis of the poultry meat production sector in the province Akhisar in Turkey. In general, the study presents some interesting data and the scientific background and design are solid. Nevertheless, there are some points in the text that need addressing for the paper to be accepted for publication.

1)      Lines 121-128: This paragraph belongs in the introduction.

2)      Lines 164-165: This sentence must go to the discussion. This is a common issue presented a lot of times in the manuscript. Since you decide to present the results and discussion separately, do not discuss the results in the RESULTS section. The same applies to lines 168-171, 180-184, and elsewhere. Please change that.

3)      Line 197: here needs a table reference (table 4).

4)      Table 4: Why aren’t any statistics in the table? You obviously compare among groups in the text but you present no statistics in the table. Please add standard errors and P values.

5)      The whole 3.2 section needs to be rewritten. There are many repetitions, statistics that are not included in the table, and an extremely detailed presentation of data that could have been in the table. Write less and present more in the respected table.

6)      Line 203: I do not understand. Re-write.

7)      Lines 231-235: you repeat the same thing twice.

8)      Lines 238-250: These data should be presented in a table because you just repeat the same things in the text twice. It was very tiring to read this paragraph.

9)      Table 5: Units are missing ($).

10)   Lines 288-298: This is a discussion, move it there.

11)   Lines 325-340: Present in one paragraph the advantages and in another the disadvantages. Don’t mix them up.

12)   Lines 347-349: That doesn’t make sense.

13)   In the discussion, there is a limited comparison of the findings of the present study with other studies and a preference for the ones that come from Turkey. Please include more references and compare your results with other studies from Europe, the US, etc., areas that are highly advanced poultry producers.

Author Response

(The authors gave the same response as above.)

Round 2

Reviewer 2 Report

I am happy to see that the authors successfully addressed the suggestions made. There is still an issue with Table 4. The f stat is not needed, discard. Give the p-value with 3 decimal places, provide the standard error for each mean, and use a,b superscripts to denote statistical differences.

Author Response

Dear Reviwer;

Our article has been revised in line with your suggestions.

Best Regards.
